# Endemic of Kazakhstan *Allium lehmannianum* Merckl. Ex Bunge and Its Position Within the Genus *Allium*

**DOI:** 10.3390/plants14071113

**Published:** 2025-04-02

**Authors:** Daulet Sh. Abdildanov, Polina V. Vesselova, Gulmira M. Kudabayeva, Bektemir B. Osmonali, Michail V. Skaptsov, Nikolai Friesen

**Affiliations:** 1Institute of Botany and Phytointroduction, 36D Timiryazev str., Almaty 050040, Kazakhstan; abdildanov00@mail.ru (D.S.A.); pol_ves@mail.ru (P.V.V.); kgm_anita@mail.ru (G.M.K.); be96ka_kz@mail.ru (B.B.O.); 2Department of Biodiversity and Bioresources, Faculty of Biology and Biotechnology, Al-Farabi Kazakh National University, Almaty 050040, Kazakhstan; 3South Siberian Botanical Garden, Faculty of Biology, Altai State University, Barnaul 656906, Russia; mr.skaptsov@mail.ru; 4Botanical Garden, School of Biology/Chemistry, Osnabruck University, 49076 Osnabruck, Germany

**Keywords:** *Allium lehmannianum*, sections *Multicaulea* and *Unicaulea*, anatomy, cytometry, nomenclature

## Abstract

The paper presents data on the phylogeny, anatomy of vegetative organs, phytocenosis composition, and ploidy of two populations of endemic *Allium lehmannianum* Merckl. ex-Bunge. in the flora of northeastern Betpakdala. Some discrepancies in nomenclature and sectional affiliation are discussed and corrected. *Allium lehmannianum* belongs to the section *Multicaulea* F. O. Khass. and & Yenglal. subgenus *Allium*, the main range covering the eastern Old Mediterranean region. The ploidy of *Allium lehmannianum* was studied for the first time using flow cytometry techniques. The sequencing of nrITS was used to construct a phylogenetic tree, including sequences from the NCBI database. The phylogenetic tree of *A. lehmannianum* was compiled by taking into account previously published data. During the fieldwork in the northeastern part of Betpakdala, the species was found to grow in a rubbly, stony, weakly undulating plain in two different communities. In this work, we applied molecular genetic, cytometric, and anatomical studies on the collected samples of two populations from the northeastern part of the Betpakdala desert to determine the taxonomic composition of a relict and endemic species of Kazakhstan.

## 1. Introduction

The uniqueness of the flora of a particular region is based on the set of endemic species that are suited to the growing conditions. In the flora of Kazakhstan, 130 onion species have been noted [1], 21 of which are endemic to the territory of the Republic [2]. At the same time, most onion species, both in general and directly endemic, are confined to mountainous areas [3]. In the desert part of Kazakhstan, three endemic species grow—*Allium lehmannianum* Merckl. ex Bunge (section *Multicaulea* F. O. Khass. and & Yenglal.), *A. iliense* Regel (section *Regeloprason* Wendelbo), A. subscabrum (Regel) R.M.Fritsch (section *Desipienta* (Omelchuk), and R.M.Fritsch subsection *Viceniprason* R.M.Fritsch and N.Friesen) [4,5]. Moreover, if the last two species belong to the group of psammophytes, then *A. lehmannianum* prefers petrophytic and clayey habitats, usually found on rubble-rocky substrates.

Here, it is necessary to note some discrepancies in the sectional assignment of *Allium lehmannianum* and the nomenclature of the sections *Multicaulea* and *Unicaulea* F.O.Khass. Initially, F. Khassanov, when describing the section *Multicaulea* F. O. Khass. and & Yenglal. [6], cited *Allium lehmannianum* as the type of species of the section. In 2011, Khassanov and his colleagues [7] described the new species *A. ravenii* and the new section *Unicaulea* with the type of species *A. kotschyi* Boiss. Aftewards Khassanov et al. [8,9] indicated that *Allium lehmannianum* belongs to the section *Unicaulea*, subgenus *Allium*, together with other species such as *A. kotschyi* Boiss and a new species for the flora of Uzbekistan, *A. ravenii* F. O. Khass., Shomur. et Kadyrov. *Allium ravenii* is morphologically close to the studied species, distinguished by a smaller flower size and the absence of bracts [7]. In the new flora of Uzbekistan [10], for the section *Multicaulea*, a new type of species, *A. borszczowii*, is given for the first time without any explanation, which is a violation of the Code of Nomenclature. The type of species is not arbitrarily changed [11]. Thus, *A. lehmannianum* belongs to the section *Multicaulea*, and the section *Unicaulea* should probably be a synonym of the section *Multicaulea*, since the type of species *A. kotschyi*, judging by the herbarium specimens viewed in GBIF https://www.gbif.org/species/2856108 (accessed on 10 January 2025) [12], is morphologically very close to *A. lehmannianum* and *A. ravenii*. In the latest work by Khassanov et al. [9], which is dedicated to both sections, the type of species for the section *Unicaulea* is already given as *A. lehmannianum* instead of *A. kotschyi* legitimately given in the section description. As for the geographical confinement of *A*. *lehmannianum*, its range is indicated differently in different sources. For example, Vvedensky in the “Flora of the USSR” [13] lists *Allium lehmannianum* (section *Porrum* Don.) for the Kyzylkum and Aral-Caspian (the species is described from the Aral deserts: Lectotype (LE 00053884): “Lehmfläche zwischen dem Jaxarts und dem See Aigiräk, 15 May 1842”, which was isolated from three sheets stored in the LE Herbarium in Sankt-Peterburg. Thus, according to Khassanov et al. [9], the information about the locus classicus of the studied species is as follows: “Clay surface between Jaxartes and Lake Aigirak Aral deserts”, which was written on the label of *A. Bunge*, and they in 1840, went to the eastern coast of the Caspian Sea, where they constantly carried out various excursions and collected rich materials and collections of fauna and flora in the Northern Aral Sea region and Mangyshlak [9]. Also, according to the expedition route and local names, Lake Aigyrak could have been located on the site of a swamp, which on maps from the mid-19th century is designated as a lake in the lower reaches of the Syr Darya River (Kazakhstan).

In the Identifier of Central Asian Plants [14], where the genus Allium was also processed by Vvedensky, the following region is already indicated: “Betpakdala, Priaralskie desert”. For the “Flora of Kazakhstan” [15], onions were processed by Pavlov and Polyakov, and in the column on the distribution of *A. lehmannianum* (section *Porrum* Don.), they indicate: Turgai, Western small hills, Kyzylorda, Betpakdala, and Kyzylkum floristic regions. At present, Khassanov et al. [9] indicated Kyzylkum, Syr Darya Karatau, Betpakdala, and Balkhash regions; however, there are no herbarium specimens in Kyzylkum, Kyzylorda region, and small western hills. Thus, it is reliably known (confirmed by herbarium specimens) (Appendix A) that *Allium lehmannianum* grows in the following regions: Turgai, Betpakdala, Shu-Ili, and Karatau (Figure 1). Interestingly, the listed territories are ancient geological formations that have undergone peneplainization [16]. The studied species is usually confined to rubble-rocky habitats and is quite rare, and despite the fairly wide representation of such habitats, it should also be noted that the height of desert hills limits the altitudinal range of the species. Combining the considered ecological and geographical characteristics allows us to classify this species as a relict element of the flora and designate its range as the eastern ancient Mediterranean [17,18]. During fieldwork in 2024 in the northeastern part of Betpakdala, this species was found in a rubble-rocky habitat.

Our work aimed to clarify the nomenclature, determine the position of the *A. lehmannianum* in the genus *Allium* system, and describe its anatomical and cytometric characteristics using molecular genetic, cytometric, and anatomical methods.

## 2. Results

### 2.1. Association of Allium lehmannianum with Plant Communities

During fieldwork in the northeastern part of the Betpakdala desert, as already noted, two populations with *A. lehmannianum* were discovered (Appendix B). In the first of them (N 46.000268 E 72.792945), this species grew in a petrophysics-perennial saltwort-wormwood community dominated by *Artemisia terrae-alba* Krasch. The community’s total projective cover (TPC) was 10–15%. The total number of species participating in forming this community is 18 (Appendix C), among which representatives of the Amaranthaceae family predominated. In the second population (N 46.600332, E 74.414978), *A. lehmannianum* was part of the stipaetum–krascheninnikoviosum community of a gently undulating plain with rare rocky hills with the dominance of Stipa caucasica Schmalh. The total number of species was also 18, most belonging to the Amaranthaceae family. The TPP of this community was 50–55%. Analyzing the species composition of these communities, it should be noted that they have common species: *Artemisia terrae-albae*, *Astragalus campylotrichus* Bunge, *Bassia prostrata* (L.) Beck, *Oreosalsola arbusculiformis* (Drobow) Sennikov, and *Tulipa biflora* Pall. (Figure 1).

In total, 30 species from 16 families were identified in the communities under consideration. The percentage ratio of the number of species of the leading families is shown in Figure 2. The quantitative predominance of the representatives of the Amaranthaceae family over the following three groups, moreover, more than twice, reflects the characteristic feature of the composition of the desert flora of Kazakhstan.

### 2.2. Anatomical Analysis

The cross-section of the *A. lehmannianum* leaf (Figure 3A) is rounded. A fairly dense cuticle layer indicates the xerophytic nature of the species. The epidermal cells (E) located in a single row under it are large in size (Table 1). Perpendicular to the epidermal cells in the centripetal direction located tightly adjacent to each other are long cells of the columnar mesophyll (PM). They are followed by small and large vascular bundles (Figure 3B) and then the main parenchyma (PX).

The stem of *A. lehmannianum* (Figure 3C) also has a rounded shape in cross-section. It should be noted that, like the leaf, the anatomical section of the stem has a dense cuticle layer. However, its spongy mesophyll (SM) cells are small (Table 2), and the outer vascular bundles and sclerenchyma cells (SC) are located around the entire circumference of the stem. At the same time, the inner vascular bundles (Figure 3D) are larger than the outer ones. The vascular bundles of the leaf and stem have a closed-collateral structure.

When studying the anatomical structure of the leaf and stem of *A. lehmannianum*, biometric measurements of the cells and their layers of two populations were performed (Table 1 and Table 2). Thus, when comparatively analyzing the biometric indicators of the leaf and stem of the studied species, it was revealed that the size of the cells of the leaf blade in the second population is larger than in the first population. In the biometric data of the stem, minor differences are observed in the two populations.

### 2.3. Flow Cytometry

The DNA content was determined in two populations of *A. lehmannianum*. The results are presented in Table 3.

The results obtained via flow cytometry showed the presence of diploids in two populations (Table 3), with an average DNA content of 35,323 pg in *A. lehmannianum*. In addition, one sample from population 2 is a probable triploid with a DNA content of 50,212 pg. Examples of uncompressed histograms of the studied *A. lehmannianum* samples are shown in Figure 4.

### 2.4. Phylogenetic Analysis

We sequenced nrITS by using five plants of *A. lehmannianum* from two populations and performed the phylogenetic analysis with all ITS sequences of the subgenus *Allium* available in the NCBI GenBank. A total of 111 sequences from the subgenus *Allium* and 6 sequences were selected as an outgroup (*A. ramosum* L., *A. tuberosum* L., and *A. trifurcatum* (F.T.Wang and Tang), J.M.Xu, *A. oreoprasum* Schrenk from the subgenus *Butomissa*, and *A. rubens* Schrad and *A. vodopjanovae N.Friesen* from the subgenus *Rhizirideum*). The alignment length of 117 ITS sequences is 691 characters; 213 characters are constant, 89 are parsimony-informative, and 389 are parsimony-informative. Unweighted parsimony analysis resulted in 2592 most parsimonious trees of 2678 steps (consistency index CI = 0.35; retention index RI = 0.79). AIC chose the substitution model HKY+G in jModelTest2 for the Bayesian analysis. The resulting phylogenetic tree (Figure 5) divides the subgenus *Allium* into three monophyletic groups, where the closest to the outgroup is the smallest clade with species of the sections *Coerulea* (Omelczuk) F.O.Khass., *Haneltii* F.O.Khass., and *Eremoprasum* (Kamelin) F.O.Khass., R.M.Fritsch, and & N.Friesen. *Allium lehmannianum* is placed in the second group, with the most significant section *Allium*. Basal is a small clade of species from the section *Avulsea* F.O.Khass., followed by species from the section *Allium* and species from the sections *Multicaulea*, *Crystallina* F.O.Khass. and Yengal., and *Brevidentia* F.O.Khass. and Yengal., which form two sister clades. All five accessions of *A. lehmannianum* (sect. *Multicaulea*) are very well separated from the neighbouring sections of *Brevidentia* and *Crystallina*. Only *A. borszszowii* (sect. *Multicaulea* after [10]) stands separately in this clade. The third group is not very strongly supported (0.97 PP and 87 BS) and consists of the sections *Mediasia* F.O.Khass., R.M.Fritsch and N.Friesen, Brevispatha Valsecchi, Pallasia (Tzag.) F.O.Khass., R.M.Fritsch and & N.Friesen, Scorodon K.Koch, Koppetdagia F.O.Khass., and partly of species from the sections *Coerulea* and *Avulsea* and from the well-supported (1.0 PP and 100 BS) species-rich section *Codonoprasum* Rchb.

### 2.5. CP Analysis

We sequenced two plastid regions by using *A. lehmannianum*: *trn*L gene, partial sequence, and the *trn*L-*trn*F intergenic spacer (*trn*L) and *psb*A-*trn*H spacer (see the NCBI GenBank accessions number in Table A2). There are few published sequences of these CP regions from the subgenus *Allium*, so we could only analyze the similarity of the sequenced CP regions using nucleotide blast analysis (https://blast.ncbi.nlm.nih.gov, accessed on 3 January 2025). Blast analysis of the CP *trn*L region from *A. lehmannianum* shows a very high similarity with *A. filidens* Regel from the sections *Chrystallina* (99.37%), *A. nikolai* F.O.Khass. (99.21%), and *A. michaelis* F.O.Khass. and Tojibaev from the section Brevidentia (99.28%), *A. ferganicum* Vved. from the section *Multicaulea* (99.28%), and *A. sabulosum* Steven ex Bunge from the section *Eremoprasa* (99.27%). Blast analysis of the CP region *psb*A-*trn*H from *A. lehmannianum* shows 100% identity with *A. nikolai* from the section *Brevidentia* and *A. filidens* from the section *Crystallina* and with many *Allium* species from the third evolutionary lineage with a higher than 99.5% similarity.

## 3. Discussion

There are not many anatomical studies of species of the subgenus *Allium*, so we cannot say much about the peculiarities of the anatomical data of *A. lehmannianum*. However, some studies [19,20] show the similarity of the anatomical structure of vegetative organs (leaf and stem) of *A. lehmannianum* for species of the subgenus *Allium*. There is a comparative difference of *A. lehmannianum* from other studies [19,20,21,22,23] in the biometric indices of leaf cells and stem anatomical structure. Examples are that in the leaf lamina’s transverse section, the vascular bundles are arranged in a circle (Figure 3A). In contrast, in the anatomical structure of the stem, the vascular bundles are arranged in two rings (Figure 3C); the outer conducting bundles are smaller than the inner one (Table 2). We attribute this to the ecological growth of the studied species.

Also, according to the database (https://cvalues.science.kew.org, accessed on 5 January 2025), in the genus *Allium*, the DNA amounts vary from a minimum of 15.20 (pg) to a maximum of 148.90 (pg). The average DNA amount in onions is 40.66 (pg). The DNA abundance of the subgenus *Allium* is poorly understood, but there are data on the DNA abundance of some species from this subgenus. For example, the amount of DNA in *Allium* caeruleum is 23.50 pg [22], while in *Allium* caesium, it is up to 25.90 pg [24,25]. Also, *Allium* sativum has 32.50 pg [22] and 49.40 pg for *Allium* filidens [26]. Our study of the average DNA content of *A. lehmannianum* in the first population was 35.323 pg and agrees with other *Allium* subgenus species. However, one sample from population 2 represents a probable triploid, so the DNA content is 50,212 pg (Table A2). Careful re-analysis of all species based on extensive material will be necessary to clarify the taxonomic value of genome size data in the subgenus *Allium*. The triploid plant found in the second population of *A. lehmannianum* indicates possible hybridization or disruption in meiosis. This population should be examined much more closely to determine the cause. Despite the finding that closely related species in the genus *Allium* can exhibit substantial differences in their DNA content, e.g., *A. coeruleum* and *A. caesium* [22,24], the genome size provides important data for taxonomic and phylogenetic analysis.

Molecular data (ITS and plastids) clearly show that *A. lehmannianum* is closely related to species from the sections *Brevidentia*, *Crystallina*, and *Eremoprasa*, and despite being a strongly under-protected clade *Allium borszszowii*, which belongs to the section *Multicaulea* [17], it is somewhat separate from all species from the sections *Brevidentia*, *Chrystallina*, and *A. lehmannianum* (Sect. *Multicaulea*). Together with other non-monophyletic sections (*Caerulea*, *Avulsea*) (Figure 5), the classification in the subgenus *Allium* still needs a lot of corrections. In addition, according to Khassanov [20], more than 70% of species from the subgenus *Allium* (375 species in the subgenus *Allium*), that have not yet been molecularly examined should be included in the analysis. Unfortunately, we could not examine other species (*A. kotschyi*, *A. ravinii*, *A. ferganicum*, *A. rinae*, *A. oxianum*) from the section *Multicaulea*. In our estimation, the sequencing of most species of the subgenus *Allium* (here, actually, the sequencing of ITS and some chloroplast fragments is sufficient) will reveal many taxonomic and phylogenetic surprises inside the subgenus *Allium*.

### Nomenclatural Notice

*Allium* sect. *Multicaulea* F.O. Khass. and Yengal., 1996 in M.A. Öztürk, Ö. Seçmen and G. Görk (eds.), Pl. Life S.W. and Central Asia 1: 148. = *Allium* sect. *Unicaulea* F.O.Khass., 2011. Stapfia 95: 174. (Typus: *Allium kotshyi* Boiis.)

Typus: *Allium lehmannianum* Merckl. ex Bunge.

## 4. Materials and Methods

The object of the study was the endemic species of Kazakhstan *A. lehmannianum* growing in the northeastern part of Betpakdala, as shown in Figure 1 and Figure 6.

### 4.1. Distribution Analyses

We compiled distribution maps from the literature and online databases and analyzed herbarium collections (LE, MW, OSBU, AA) and herbarium acronyms according to the Index Herbariorum [27].

Classical botanical (route-reconnaissance, ecological-systematic, ecological-geographical) methods were used in the research. The following fundamental summaries were used to identify the collected material: “Flora of Kazakhstan” [15], “Illustrated Guide to Plants of Kazakhstan” [28], “Guide to Plants of Central Asia and Kazakhstan” [14] and works by authors studying the genus *Allium* L. [29,30,31,32]. The names of plant species were given by the data from the Plants of the World website [33]. The authorship of species, genera, and families has been critically cross-checked against the information in the International Plant Names Index [34]. The material from the Plantarium website [35] was also used. The QGIS 3.40 program was used for data mapping (https://qgis.org, accessed on 5 January 2025).

### 4.2. Anatomical Analysis

The specimens for the anatomical studies were collected in vivo during the flowering phase from two populations of the study species located in the western and northern parts of the Betpakdala desert (Central North Turanian subprovince of the North Turanian province of the Iranian-Turanian subregion of the Sahara-Gobi desert region) [36]. Objects for anatomical sections were fixed in 70% alcohol. For the office processing of specimens, 15 × 15 mm histological paraffin was used for fixation. Materials were stained in methylene blue eosin according to May–Grunwald. Cross-sections of samples were made using the automatic microtome (MEDITE M530, MEDITE, Burgdorf, Germany) [37]. Transverse slices were obtained by employing a Levenhuk D740T 5.1 camera using Levenhuk Lite software at ×720, ×1000 magnification. Biometric data were also obtained using this software. The mean number and standard error of biometric data were calculated using Microsoft Excel software using the data analysis function.

### 4.3. Flow Cytometry

The DNA content was determined via flow cytometry techniques with propidium iodide (PI) staining. Leaves dried with silica gel were used as samples. Samples were chopped using a standard sharp razor blade in LB01 buffer containing PI (50 µg/mL), RNase (50 µg/mL) [38,39], and supplemented with 12 mM sodium thiosulfate and 1% polyvinylpyrrolidone [38]. The nuclear suspension was filtered through a nylon filter with a pore size of 30 μm. Analyses were performed on a Cytoflex cytometer (Beckman Coulter, Inc.) with a 488 nm laser and 610 excitation filter. Analyses were performed on a Cytoflex (Beckman Coulter, Inc., Brea, CA, USA) cytometer. Peaks with at least 1000 nuclei and a CV of less than 5% were analyzed. Histograms were visualized and processed using CytExpert software (Beckman Coulter, Inc., Brea, CA, USA). Descriptive statistics were calculated using XLStat (Addinsoft, Paris, France). An internal standard was used: the Pisum sativum ‘Ctirad’, 2C = 9.09 pg and Vicia faba ‘Inovec’, 2C = 26.9 pg [38,39]. The DNA content was calculated from the ratio of internal standard to sample peaks. The expected ploidy level was interpreted based on published DNA content data in the Kew C-value database for closely related species [40,41]. The angiosperm DNA C-values database (release 9.0, April 2019) (https://cvalues.science.kew.org/) [41] was referred to.

### 4.4. DNA Extraction, Amplification, and Sequencing

Plant material for DNA extraction was isolated from silica dried leaves collected in the field. Newly sequenced accessions were marked with Am numbers in the trees, and their origin is shown in Table A3. Sequences from the NCBI GenBank are marked with GenBank accession numbers. The total genomic DNA was isolated from silica dried leaves collected in the field using the Diamond Plant DNA Kit (http://diamond-dna.ru/) according to the manufacturer’s instructions and used directly in the PCR amplification. The complete nuclear ribosomal ITS region (ITS1, 5.8S, and ITS2) was amplified using the primers ITS-A [42] and ITS-4 [43]. The chloroplast regions *psb*A-*trn*H were amplified using the primers published by Shaw et al. [44] and the *trn*L intron und *trn*L - *trn*F spacer was amplified using the primers published by Taberlet et al. [45]. PCR products were sent for sequencing to Microsynth SeqLab (Göttingen, Germany; www.microsynth.seqlab.de). The sequences from all the individuals were manually edited in Chromas Lite 2.1 (Technelysium Pty Ltd., South Brisbane, Australia) and aligned with ClustalX [46], and the alignment was manually corrected using MEGA 7 [47].

### 4.5. Phylogenetic Analyses

Sequences from *Allium ramosum*, *A. tuberosum*, *A. trifurcatum*, and *A. oreoprasum* (*A. subgen*. Butomissa) and *A. vodopjanovae N.Friesen* and *A. rubens* (*A. subgen*. Rhizirideum) from the third evolution lineage [29,48] were chosen as the outgroup. ITS data sets were analyzed for position identification in the third evolutionary lineage and to find the relationships of *A. lehmannianum* through parsimony (PAUP) and Bayesian phylogenetic analysis (MrBayes). Fitch parsimony was determined via the heuristic search option in PAUP version 4.0b10 [49] with MULTREES, TBR branch swapping, and 100 replicates of random addition sequences. Gaps were treated as missing data. The consistency index (CI) [50] was calculated to estimate the amount of homoplasy in the character set. The most parsimonious trees returned by the analysis were summarized in one consensus tree using the strict consensus method. Bootstrap analyses using 1000 pseudoreplicates were performed to assess the support (BS) of the clades [51]. Bayesian phylogenetic analyses were also performed using MrBayes 3.1.23 [52]. The sequence evolution model was chosen following the Akaike Information Criterion (AIC) obtained from jModelTest2 [53]. Two independent analyses with four Markov chains were run for 10 million generations, sampling trees every 100 generations. The first 25% of trees were discarded as burn-in. The remaining trees were combined into a single data set, and a majority-rule consensus tree was obtained along with posterior probabilities (PP).

## 5. Conclusions

The botanical descriptions of both phytocenoses with the participation of *Allium lehmannianum* showed up to 30 plant species from 16 families. Five species were found in both populations (*Artemisia terrae-albae*, *Astragalus campylotrichus*, *Bassia prostrata*, *Oreosalsola arbusculiformis*, *Tulipa biflora*).

The examination of the species via flow cytometry revealed triploidy in the second population, but the DNA amounts of some species of the subgenus *Allium* have nearly the same amount of DNA. The occurrence of triploid specimens can be explained by possible hybridization.

The analysis of the peculiarities of the anatomy of the vegetative organs of *A. lehmannianum* showed the derived structures of leaf and stem such as the presence of a thickened outer wall of the epidermis in the leaf; conductive bundles are arranged in a circle in the form of the closed collateral type; and in the anatomical structure of the stem there are internal and external conductive bundles. The main difference between the two populations is the biometric indices of the vegetative organs of *A. lehmannianum*.

According to the modern phylogenetic tree, *Allium lehmannianum* belongs to the section *Multicaulea*, and the section *Unicaulea* should probably be a synonym of the section *Multicaulea*. Further studies of the subgenus *Allium*, which have not yet been studied molecularly, are also needed.

## Figures and Tables

**Figure 1 plants-14-01113-f001:**
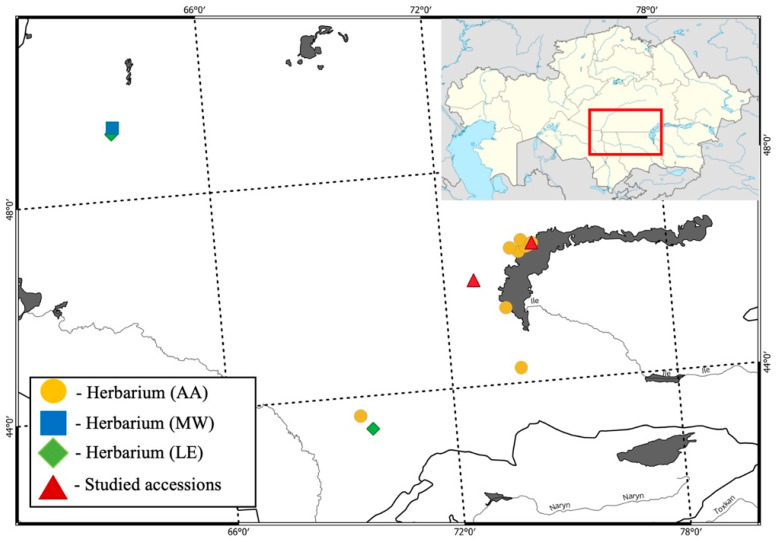
Distribution of *A. lehmannianum* according to our collections and herbarium data.

**Figure 2 plants-14-01113-f002:**
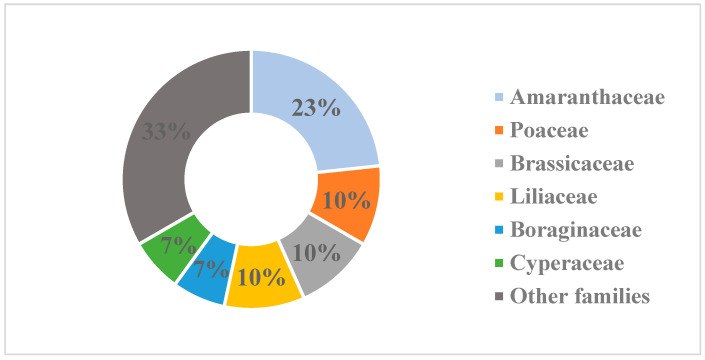
Family spectrum of species composition of two communities with participation of *A. lehmannianum*.

**Figure 3 plants-14-01113-f003:**
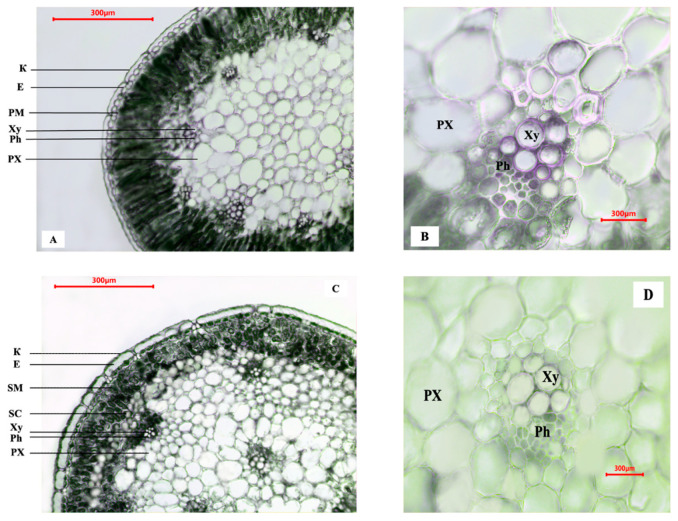
Comparative anatomical structure of the studied species: (**A**) transverse section of the leaf plate (×720), (**B**) structure of the conductive bundle of the leaf plate (×1000), (**C**) cross-section of the stem (×720), (**D**) structure of the conductive bundle of the stem (×1000). K—cuticle, E—epidermis, PM—polysaccharide mesophyll, SM—spongy mesophyll, SC—sclerenchyma, Xy—xylem (outer), Ph—phloem (outer), PX—main parenchyma.

**Figure 4 plants-14-01113-f004:**
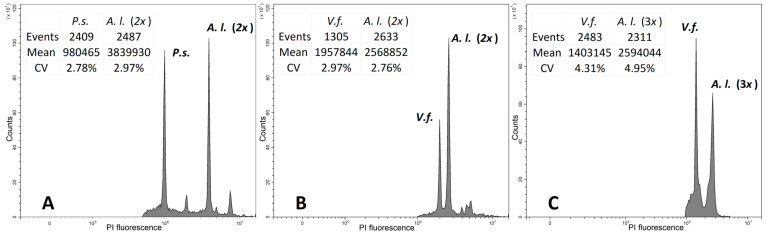
Examples of flow cytometric histograms of the *A. lehmannianum* samples (log scale). (**A**) diploid cytotype with *P. sativum* as internal standard; (**B**) diploid cytotype with *V. faba* as internal standard; (**C**) triploid cytotype with *V. faba* as internal standard.

**Figure 5 plants-14-01113-f005:**
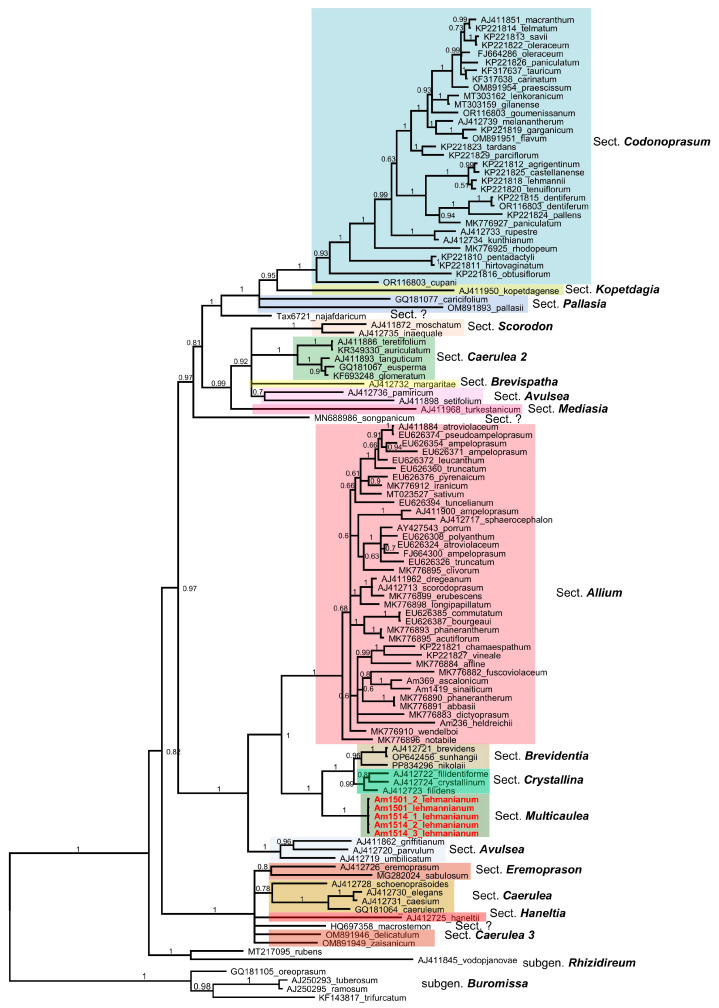
Phylogenetic nrITS tree of the subgenus *Allium* of genus *Allium*. Numbers by nodes represent bootstrap support (1000 replicates) and Bayesian probabilities. The joint presence of Bayesian probabilities over 0.98 and bootstrap support over 95% is indicated with a black dot. For the origin of samples without GenBank accession numbers, see Table A2.

**Figure 6 plants-14-01113-f006:**
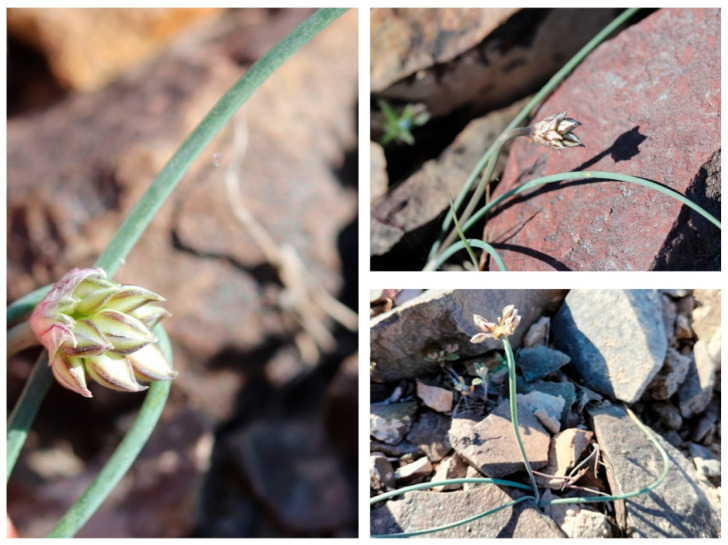
Object of study *A. lehmannianum*.

**Table 1 plants-14-01113-t001:** Biometric data of anatomical sections of leaf blade of *A. lehmannianum* (μm).

K	E	PM	Xy	Ph	PX
Population 1					
21 ± 3.36	21.7 ± 3.39	79.85 ± 9.17 × 16.21 ± 2.30	8.80 ± 1.46	8.65 ± 2.25	42.36 ± 4.13
Population 2					
21 ± 3.02	23.30 ± 3.05	90.23 ± 9.51 × 20 ± 2.8	10.27 ± 2.27	5.82 ± 0.95	58.74 ± 4.44

**Table 2 plants-14-01113-t002:** Biometric data of anatomical sections of the stem of *A. lehmannianum* (μm).

K	E	SM	SC	Xy	Ph	Xy_2_	Ph_2_	PX
Population 1								
18 ± 3.36	24.19 ± 3.45	23.35 ± 3.09	24.4 ± 2.04	8.97 ± 1.55	4.76 ± 1.46	12.89 ± 4.15	9.04 ± 3.36	51.18 ± 4.45
Population 2								
21 ± 3.02	19.55 ± 3.16	21.21 ± 4.19	19.59 ± 5.34	9.22 ± 1.55	7.59 ± 1.89	18.22 ± 3.44	12.74 ± 2.14	51.27 ± 4.43

Legend: K—cuticle, E—epidermis, SM—spongy mesophyll, SC—sclerenchyma, Xy—xylem (outer), Ph—phloem (outer), Xy_2_—xylem (inner), Ph_2_—phloem (inner), PX—main parenchyma.

**Table 3 plants-14-01113-t003:** DNA content of the studied *A. lehmannianum* samples.

Species	Locality	Mean 2C, pg	SD, pg	CV	1C, Gbp	Expected Ploidy
*A. lehmannianum* (2×)	Pop 1	35.468	0.405	1.14%	34.688	2
*A. lehmannianum* (2×)	Pop 2	35.104	0.292	0.83%	34.332	2
Mean 2C for diploid cytotype		35.323	0.393	1.11%		
*A. lehmannianum* (3×)	Pop 2	50.212	0.417	0.83%	49.107	3

## Data Availability

All data supporting this study’s findings are available in the main text or Appedixes.

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
