# Peer review of "Endemic of Kazakhstan Allium lehmannianum Merckl. Ex Bunge and Its Position Within the Genus Allium"

_plants, 2025, doi:10.3390/plants14071113_

Round 1

Reviewer 1 Report

Comments and Suggestions for Authors

This study explores Allium lehmannianum in the Betpakdala desert, examining its ecological, anatomical, and molecular features. It identifies 30 plant species in its habitat, reveals triploidy in one population through flow cytometry, and places the species in the Multicaulea section. The study emphasizes the need for further molecular research on the subgenus Allium.

Abstract

The abstract could be more concise, improving the flow by combining related information on species and habitat.

Highlighting the key findings or results of the study would strengthen the abstract and engage the reader.

Introduction

The discussion on the nomenclature and sectional classification of Allium lehmannianum could be made clearer by summarizing the key points of disagreement more concisely.

The ecological and geographical significance of A. lehmannianum should be more clearly linked to its classification as a relict species for better understanding.

Discussion

The anatomical comparisons to other species in the subgenus Allium could be expanded, providing more detail on how the anatomical characteristics of A. lehmannianum specifically compare to other closely related species.

The discussion on DNA content and its taxonomic implications could benefit from a more in-depth exploration of how genome size variations might influence the classification of A. lehmannianum within the subgenus.

The mention of unexamined species from the subgenus Allium highlights a limitation; it would be helpful to discuss the potential impact of including these species in future molecular analyses for refining the classification system.

Methods

The section describing the anatomical analysis could be more concise by summarizing the key techniques and methods used for tissue fixation, staining, and sectioning, rather than listing all steps in detail. A clearer focus on the most relevant aspects would improve readability.

The flow cytometry section would benefit from consistent terminology, especially regarding the use of internal standards and the description of the cytometer model. A clearer explanation of how the flow cytometry data contributes to the study's conclusions would also help reinforce its importance.

Conclusion

The conclusion could be more focused by clearly highlighting the key species identified in the study, rather than mentioning them all in a long list, for better readability.

The triploidy finding in the second population is mentioned, but a more detailed discussion on its implications for the taxonomy and genetics of A. lehmannianum would strengthen the conclusion.

The need for further molecular studies of the subgenus Allium is noted, but specifying what additional molecular data or techniques could be useful would make the recommendation more actionable.

Author Response

Answer to reviewer
Reviewer 1. We are grateful for the comments and suggestions of the first reviewers, which make the manuscript much better.
1. The abstract could be more concise, improving the flow by combining related information on species and habitat. Highlighting the key findings or results of the study would strengthen the abstract and engage the reader.

Answer: We have rewritten the abstract to be more precise
2. Introduction: The discussion on the nomenclature and sectional classification of Allium lehmannianum could be made clearer by summarizing the key points of disagreement more concisely.
Answer: In more detail, we have described and supplemented the nomenclatural discrepancies between the Unicaulea and Multicaulea sections.
3. The ecological and geographical significance of A. lehmannianum should be more clearly linked to its classification as a relict species for better understanding.
Answer: We believe that the ecological and geographical description is clear enough

4. Discussion:
The anatomical comparisons to other species in the subgenus Allium could be expanded, providing more detail on how the anatomical characteristics of A. lehmannianum specifically compare to other closely related species.
Answer: We have supplemented the comparison of anatomical data with additional publications.
5. The discussion on DNA content and its taxonomic implications could benefit from a more in-depth exploration of how genome size variations might influence the classification of A. lehmannianum within the subgenus.
Answer: Not many Allium species from the subgenus Allium have been examined for DNA content so far to speak of any significance for classification. DNA content is very important information in describing a species.
6. The mention of unexamined species from the subgenus Allium highlights a limitation; it would be helpful to discuss the potential impact of including these species in future molecular analyses for refining the classification system.
Answer. Molecular analysis of all species from the subgenus Allium will bring many new insights into phylogeny and classification. All you need is to sequence ITS and some chloroplast fragments.
Methods: The section describing the anatomical analysis could be more concise by summarizing the key techniques and methods used for tissue fixation, staining, and sectioning rather than listing all steps in detail. A clearer focus on the most relevant aspects would improve readability.
Answer. We have rewritten the anatomical part.
The flow cytometry section would benefit from consistent terminology, especially regarding the use of internal standards and the description of the cytometer model. A clearer explanation of how the flow cytometry data contributes to the study's conclusions would also help reinforce its importance.
Answer: We have not found any discrepancies in terminology, but the description of the cytometry model has been supplemented
Conclusion:
The conclusion could be more focused by clearly highlighting the key species identified in the study, rather than mentioning them all in a long list, for better readability.
Answer: We have rewritten the first part.
The triploidy finding in the second population is mentioned, but a more detailed discussion on its implications for the taxonomy and genetics of A. lehmannianum would strengthen the conclusion.
Answer: For taxonomy, first, there will be no implication, but a possible explanation for the triploidy is the hybridisation.
The need for further molecular studies of the subgenus Allium is noted, but specifying what additional molecular data or techniques could be useful would make the recommendation more actionable.
Answer: All you need is to sequence ITS and some chloroplast fragments. In the future, complete genome sequencing might be possible, but then for all of the more than a thousand Allium species.

Reviewer 2 Report

Comments and Suggestions for Authors

The article is interesting and examines Allium lehmannianum from various points of view, from the taxonomic framework to phylogeny, ploidy etc. I note numerous typos that I have highlighted in the attached file. the organization of the paper could be improved, for example, figure 1 should be cited in the introduction and not in the following paragraph, as it is in the introduction that the distribution of this species is discussed. the paragraph relating to the plant community where this species grows is discussed at the beginning of the results but is less relevant than the rest of the results, which mainly concern taxonomic and morpho-anatomical aspects of the species. Therefore I believe it could be moved to the last paragraph of the results. With respect to the sections to which Allium lehmannianum refers, the authors do not take a firm position and the picture is still unclear, but I respect this decision, although a clearer proposal could be made on the basis of the knowledge achieved so far.
The bibliography should also be carefully revised in formatting.
For the rest, after a good revision, the article could be published on Plants

Author Response

Reviewer 2. We are grateful for the comments and corrections of the second reviewers, which make the manuscript much better. He read the text very carefully and corrected many of our mistakes. We have adopted all his suggestions.

Bibliography was carefully revised in formatting

Round 2

Reviewer 1 Report

Comments and Suggestions for Authors

Accept

Reviewer 2 Report

Comments and Suggestions for Authors

The already interesting and well-written article has been significantly improved by the authors and can therefore be published on Plants

 Note: On line 300 add a square bracket after the citation 39